# What Is the Relationship Between Efficacy of Seed Treatment with Insecticides Against *Dalbulus maidis* (Delong and Wolcott) (Hemiptera: Cicadellidae) Healthy and Infected with Spiroplasm in the Corn Stunt Control?

**DOI:** 10.3390/insects16070713

**Published:** 2025-07-11

**Authors:** Ana Carolina M. Redoan, Vinicius M. Marques, Poliana S. Pereira, Ivênio R. de Oliveira, Dagma D. Silva-Araújo, Luciano V. Cota, Marcos Antonio M. Fadini, Charles M. Oliveira, Diego D. Rafael, Simone Mendes

**Affiliations:** 1Brazilian Agricultural Research Corporation (Embrapa), Sete Lagoas 35702-098, Brazil; ac.redoan@gmail.com (A.C.M.R.); ivenio.rubens@embrapa.br (I.R.d.O.); dagma.silva@embrapa.br (D.D.S.-A.); charles.oliveira@embrapa.br (C.M.O.);; 2Department of Biosystems Engineering, Federal University of São João del Rei (UFSJ), São João del Rei 36307-352, Brazil

**Keywords:** chemical control, leafhopper, integrated management, spiroplasma, *Zea mays*

## Abstract

The corn leafhopper *Dalbulus maidis* is an insect vector that inoculates phytopathogens into the phloem, compromising nutrition, growth, and corn development. Symptoms of corn stunt include chlorosis, shortened internodes, plant malformation, ear proliferation, and even plant mortality. In more severe cases, it can reduce corn yield by up to 100%. We evaluated the effect of the main insecticides for seed treatment on the control of infective and non-infective leafhoppers, the persistence of the treatment’s effect, and its relationship to the expression of disease symptoms and yield. We observed that seed treatment was effective until the V2 corn stage and that infective leafhoppers were more sensitive to insecticides compared to non-infective ones. Plants were more susceptible to the disease in the early vegetative stages, significantly reducing corn yield.

## 1. Introduction

The corn leafhopper *Dalbulus maidis* (Delong and Wolcott) (Hemiptera: Cicadellidae) is the main pest currently affecting corn in Brazil and several countries in Latin America [1,2]. This insect is the vector of the bacteria (Mollicute class), spiroplasma (corn stunt spiroplasma—CSS), phytoplasma (maize bushy stunt phytoplasma—MBSP), and viruses (virus maize rayado fino virus (MRFV) and maize striate mosaic virus (MSMV)) associated with the corn stunting disease complex (CSDC) [3,4]. The *D. maidis* feeds on phloem sap and transmits phytopathogens in a persistent and propagative manner, except for MSMV that transmits in a persistent circulative manner, and these phytopathogens occur within the same geographical distribution [5]. CSDC directly impacts the development of maize plants, potentially causing losses of up to 100% of the yield. To manage the insect vector and related diseases, a set of good agricultural practices, primarily preventive, is recommended to reduce the risk of high CSDC incidence [2,6].

Among the pathogens involved in CSDC, phytoplasma and spiroplasma have been considered the most important in terms of negative impacts on grain production [7]. These pathogens can influence the development of *D. maidis* either positively or negatively, although there is no consensus in the results of studies conducted with these pathogens. For example, while some studies suggest that CSS increases the survival of its vector when exposed to temperatures between 10 and 20 °C [8], but does not affect adult leafhopper survival compared to uninfected ones when reared at 26 °C [5], another study found that both CSS and MBSP enhanced the survival of *D. maidis* at cool (15 °C) and warmer (31 °C) temperature conditions [9]. The effects of pathogens on the development of *D. maidis* s have not yet been fully described, and the lack of this knowledge can directly influence management strategies.

Because of the slow development of these pathogens in the vascular systems of maize plants, symptoms of CSDC are usually observed during the grain production stage. The earlier the plants are infected, the greater the losses are [7]. Therefore, management measures for the insect vector should be implemented in the early stages of the crop, particularly between emergence and the V8 stage [6]. In this context, the use of organosynthetic insecticides for seed treatment can be essential in delaying the transmission of pathogens by *D. maidis*, thereby reducing the pathogen load inoculated by the vector in the plants [10,11,12,13]. Chemical control through seed treatment is a common tactic that can provide from 10 to 15 days of protection against *D. maidis* [10,13]. In Brazil, there are 74 commercial insecticides registered for controlling *D. maidis*, consisting of combinations of 13 active ingredients from six chemical groups and only five modes of action [14,15]. These products are neuromuscular disruptors, acting on both cholinergic synapses and ion channels in insect neuron axons [16]. However, there are no studies that have demonstrated the efficacy of a seed treatment to control *D. maidis* and to reduce the damage caused by stunting

*D. maidis* has caused significant economic losses in maize-producing regions, particularly in Argentina, Brazil, and Mexico. In Brazil, the first significative outbreak led to an 85% increase in insecticide usage. In Argentina and Mexico, yield reductions range from 30% to 50% [1,6]. Given that the presence of pathogens (phytoplasma and spiroplasma) can influence the characteristics and behavior of *D. maidis*, it is possible that biotic and abiotic factors, including the application of chemical insecticides, may produce varying results based on the infection status of the insects. The interaction of these chemical products with both healthy and infected insects has not been explored, and understanding this relationship would contribute to a deeper knowledge of this pathosystem. This study aimed to evaluate the effect of insecticide treatment of maize seeds on the mortality of both infected with spiroplasma and healthy adult *D. maidis*, and to investigate how this treatment influences the expression of stunting symptoms and its relationship with reductions in productivity.

## 2. Materials and Methods

### 2.1. Insects

The experiment was conducted at the Embrapa Maize and Sorghum Research Center in Sete Lagoas, Minas Gerais, Brazil. We used the method described by Oliveira and Sabato (2017) [7] to obtain the infectious leafhoppers. For five consecutive weeks, 50-day-old maize plants (*Zea mays* L., cultivar LP2020) infected with spiroplasma, grown in pots with 5 kg of soil, were used as the source of infection. Approximately 500 healthy adult *D. maidis* were placed on each infected plant, enclosed in a voile bag tied to the plant stem. After five days in maize source plants (CSS), the leafhoppers were transferred to cages with healthy maize seedlings. After 23 days (latent period), the infective *D. maidis* adults were used to infest the maize plants in the experiments as follows.

### 2.2. Maize Seeds, Insecticides, and Experiments

The maize hybrid provided by Santa Helena Sementes in Cruz Alta, Brazil (SHS7970 PRO3) was used in the experiments. The insecticides, their active ingredients and application rates are shown in Table 1. The experiments were conducted from September to December 2023 in Sete Lagoas, Minas Gerais, Brazil.

The maize seeds were treated with insecticides and sown on the same day in 200 mL plastic pots filled with fertilized soil, using 3 seeds per pot. After emergence, thinning was performed, leaving one plant per pot. Fertilizers NPK 8-28-16 (Fertilizantes Heringer, Iguatama, Brazil), limestone, and micronutrients (FTE) were used for plant maintenance and growth. Each plant was infested only once, and the experiment was conducted separately for healthy and infective leafhoppers.

In total, there were five infestations (one per time), corresponding to 0, 5, 10, 15, and 20 days after the appearance of the second expanded leaf, which corresponded to the maize’s phenological growth stages: V2, V3-V4, V4-V5, V5, and V5-V6. The vegetative stages from V2 to V6 in maize represent key phases in early plant development, each marked by the number of fully visible leaf collars [17]. On each infestation date, seven leafhoppers (infected with CSS or healthy, separately) were placed inside polyethylene terephthalate (PET) bottles, whose upper part was closed with voile fabric, containing a plastic pot with one maize plant. Insect mortality was evaluated for five consecutive days (24, 48, 72, 96, and 120 h). The surviving insects at the end of the fifth day of evaluation were removed and discarded. The design was completely randomized (CRD) with 5 infestation times, 7 insecticide seed treatments (6 insecticides + water), and 2 groups of leafhoppers (infected with CSS and healthy), with 5 replicates corresponded to one plant per pot. Totaling 350 plants, with 175 pots containing plants exposed to healthy leafhoppers and 175 pots containing plants exposed to infective leafhoppers. A total of 2.450 adult *D. maidis* were used (Figure 1).

After the leafhopper mortality evaluation period, maize plants exposed to leafhoppers were transplanted to pots (26 cm diameter and 29 cm high) containing soil fertilized as recommended for maize cultivation. Plants were kept in a greenhouse, where CSS symptoms, plant height (determined from the ground level to the tassel insertion node), and the number and size of ears per plant were measured with a tape measure.

The severity of spiroplasm was assessed in the reproductive maize growth stage (R4), when the consistency of the kernel interior is similar to “dough” [17] We used a symptom rating scale ranging from 1 to 6: 1—no symptoms (health plant), 2—plant with symptoms (reddening or yellowing) with normal height, 3—plant with symptoms and reduction in size, 4—plant with symptoms and early drying, 5—plant with symptoms, reduction in size and early drying, and 6—plant with symptoms severe reduction in size and falling over [18].

After the visual assessments of CSS symptoms, molecular analyses were performed using polymerase chain reaction (PCR) to confirm the infection. Samples were collected 90 days after inoculation, placed in identified falcon tubes and stored in a freezer at −60 °C. DNA extraction for maize plants followed the protocol proposed by Saghai-Maroof et al. (1984) [19] with the modifications suggested by Sousa and Barros (2017) [20]. The PCR test was performed using PCR-based methods as described by Sousa and Barros (2017) [20], using primers CSSF2 (5′-GGC AAA AGA TGT AAC AAA AGT-3′) and CSSR6 (5′-GTT ACT TCA ACA GTA GTT GCG-3′) for the detection of *S. kunkelii* [21].

### 2.3. Statistical Analysis

For preliminary data analysis, the Shapiro–Wilk (0.05) and Levene (0.05) tests were used to assess normality and homogeneity of variance. The presence of discrepant values (=“outliers”) was also assessed through direct observation in box-plot graphs. The R statistical environment version 4.4.1 (R Core Team 2024) [22] was used to adjust the models and generate the graphs. The ‘MASS’, ‘car’, and ‘rstatix’ packages were used for the analyses to adjust the models, and the ‘ggplot2’ package was used to generate the graphs. The numerical database, as well as the scripts for adjusting the models and generating the graphs, were stored in an open GitHub address to increase the checking and reproducibility of the results [23].

#### Analysis of Mortality of Healthy and Infective Leafhoppers

Generalized Linear Models (GLM) with a binomial distribution were used. This type of distribution is suitable for ratio data that have low variation at extreme values and high variation at intermediate values [23]. Initially, the full model was adjusted, with the explanatory variables being insecticide seed treatment, time of day, and leafhopper infectivity (infective or healthy), as well as the interactions between these variables. The adjusted models were evaluated for overdispersion, the Akaike criterion (AIC), and the random distribution of the residuals. If any terms were found to be insignificant, they were removed, and the models were readjusted. When necessary, the quasibinomial distribution was used to reduce overdispersion [23].

Since daily mortality (%) is a non-normally distributed continuous variable without homogeneity of variance, the Kruskall–Wallis nonparametric test was performed to assess the differences among groups. Data were tested considering different insecticide treatments (imidacloprid/Tiodicarb, λ-cialotrin/Thiametoxam, imidacloprid, Clothinidin, Thiametoxam, and T. methyl + fipronil/Thiametoxam), different times of evaluation (24 h, 48 h, 72 h, 96 h, and 120 h), and if the leafhopper was infective or healthy. The significance threshold value was set at *p* < 0.05. When there were differences, a Mann–Whitney (U test) was performed to evaluate whether the mortality (%) differed between the two groups (post-hoc test).

The study used simple linear regression to analyze the relationship between maize plant height, ear length, and stunting score at a 5% significance level. The Kruskal–Wallis non-parametric test was employed to assess the rank values of stunting scores in relation to the type of insecticide seed treatment, as the score data did not have a defined frequency distribution. Dunn’s non-parametric post-hoc test at 5% significance was used to further evaluate the blight score readings taken at the time of planting (5 days after seed treatment).

## 3. Results

Insecticide seed treatment significantly affected the mortality of *D. maidis* in comparison to the control group (GLM: χ^2^ = 72.98, df = 6, *p* < 0.001) (Figure 2A). In addition, there was a significant difference in mortality between healthy and infective leafhoppers, with the spiroplasma-infected leafhoppers showing higher mortality compared to the healthy leafhoppers (GLM: χ^2^ = 61.98, df = 6, *p* < 0.001) (Figure 2B).

The interactions among the insecticides seed treatments, leafhopper infectivity, and evaluation time after seed treatment were significant (Appendix A). *D. maidis* mortality was affected by individual infectivity (χ^2^ = 115.19, df = 1, *p* < 0.001), seed treatment (GLM: χ^2^ = 93.45, df = 6, *p* < 0.001), and time after treatment (GLM: χ^2^ = 90.49, df = 1, *p* < 0.001). For the control treatment (water), neither time (GLM: χ^2^ = 2.77, df = 1, *p* = 0.096) nor leafhopper infectivity (GLM: χ^2^ = 1.22, df = 1, *p* = 0.268) affected *D. maidis* mortality (Figure 3A).

For all insecticides used in seed treatments, a higher mortality was observed for both infectious and healthy leafhoppers between time 0 and 5 days after seed treatments. As we can see in the results for seed treatments, both time and infectivity had a significant effect on *D. maidis* mortality, respectively: imidacloprid/thiodicarb (χ^2^ = 12.89, df = 1, *p* < 0.001) and (χ^2^ = 20.80, df = 1, *p* < 0.001) (Figure 3B); Lambda-Cyhalothrin/thiamethoxam (χ^2^ = 12.89, df = 1, *p* < 0.001) and (χ^2^ = 20.80, df = 1, *p* < 0.001) (Figure 3C); imidacloprid (χ^2^ = 9.14, df = 1, *p* = 0.004) and (χ^2^ = 9.15, df = 1, *p* = 0.002) (Figure 3D); Clothianidin (χ^2^ = 10.94, df = 1, *p* < 0.001) and (χ^2^ = 35.59, df = 1, *p* < 0.001) (Figure 3E); thiamethoxam (χ^2^ = 31.26, df = 1, *p* < 0.001) and (χ^2^ = 59.36, df = 1, *p* < 0.001) (Figure 3F); and thiophanate-methyl + fipronil/thiamethoxam (χ^2^ = 30.21, df = 1, *p* < 0.001) and (χ^2^ = 31.21, df = 1, *p* < 0.001) (Figure 3G).

When we evaluated *D. maidis*’ daily mortality, the insecticide imidacloprid/thiodicarb showed the highest mortality: 74% within the first 24 h and 100% at 72 h for healthy leafhoppers, and over 80% mortality for infective ones between 24 and 120 h from time zero. This treatment was the most effective in preventing the insect from remaining in contact with the plant, thereby stopping feeding and the transmission of the pathogen (Figure 4).

Thiamethoxam and thiophanate-methyl + fipronil/thiamethoxam showed high mortality after 72 h for infective leafhoppers, whereas for healthy leafhoppers, the mortality was below 50%. The seed treatments were efficient, negatively affecting the leafhoppers only at time zero. In other words, the treatment protected the plant for up to 5 days after the emergence of the second fully expanded leaf (V2), or for 11 days after planting (Figure 4).

Visual assessment revealed the impact of leafhoppers infected with S. kunkelli on corn plants (Kruskal–Wallis test: χ^2^ = 23.08, df = 6, *p* < 0.001) at time zero, the only case that had statistically significant results (Figure 5). Imidacloprid/thiodicarb treatment demonstrated efficacy in achieving high mortality rates of insects in a short period, which reflected in the lowest stunt score of 1 (no symptoms) (Dunn Test: *p* < 0.001) (Figure 5). Molecular analyses of the plants confined with the infective leafhoppers at time zero confirmed the results of the scores assigned through visual assessment (Appendix A).

Plants without spiroplasm symptoms had greater vigor and yield than those infested. Both plant height and yield decreased according to disease severity, as indicated by the stunting score (Figure 6). Plants without symptoms (score 1) reached a height of 2.0 m and had ear lengths of 15 cm. In contrast, plants showing symptoms, reduced height, and lodging (score 6) had heights ranging from 1.0 to 1.5 m, with ear lengths below 10 cm.

## 4. Discussion

This result is significant due to the complexity and low efficiency of *D. maidis* and CSDC control in the field. There is a gap in our understanding regarding the bioecology of *D. maidis* and its interactions with the pathogen, resulting in a lack of knowledge about variations in population occurrence in initial growing stages [24] and CSDC transmission. Also, further experiments should be carried out to elucidate questions about the resistance of populations to synthetic organic insecticides.

Given the leafhopper’s migratory behavior, it is crucial to synchronize sowing and avoid staggered planting to prevent the presence of corn plants at varying growth stages [25]. Maize crops at different growth stages in nearby areas allow the overlapping of the plant’s life cycle. This dynamic favors the multiplication and migration of leafhoppers from mature crops to new ones in the early development stages, efficiently carrying and transmitting mollicutes to these young plants [26].

The use of insecticides has been one of the main strategies for controlling *D. maidis* and CSDC in the field [6,13,27]. Even so, studies published to date on the efficacy of seed treatments for controlling *D. maidis* have prioritized the action of insecticides only on infective leafhoppers [24,27,28,29,30]. Thus, the differential in mortality of healthy and infective leafhoppers is very significant, and the contributions of this information to pest management are addressed in the present study. In this context, the results demonstrate that the damage caused by corn stunt disease is not proportional to the size of the vector population but rather to its infectivity rate [2].

Oliveira et al. (2008) [13] verified the efficiency of corn seed treatment for controlling *D. maidis*. In this case, seeds were treated with imidacloprid and thiamethoxam, and healthy leafhoppers were confined with infective ones. These authors found a control efficiency for *D. maidis* adults equal to or greater than 70% up to 30 days after plant emergence, requiring between 4 and 24 h to achieve these mortality rates. Ruegger (2019) [31] evaluated *D. maidis* adult mortality in corn (V3 and V4) via seed treatment with the insecticides chlorantraniliprole, thiamethoxam, and imidacloprid. Only thiamethoxam and imidacloprid demonstrated high efficacy, with mortality rates exceeding 80% on leafhopper populations, reducing their feeding rates. Neves et al. (2021) [28] showed that seed treatment remained effective until the V4 stage when neonicotinoids were used, lowering the disease score (between 2 and 3) for corn stunt and reducing yield losses by 20% to 60%. These results align with those found in this study, where the residual effect of insecticides typically lasted less than three weeks.

Our results showed that infective *D. maidis* adults carrying spiroplasma exhibited higher mortality when exposed to insecticide-treated seeds compared to healthy adults (Figure 2B). This suggests that pathogen infection makes *D. maidis* individuals more susceptible to the action of insecticidal compounds. During the mollicute development, muscle tissues in the body of the leafhopper are damaged [32,33]. As the muscle cells of the midgut and Malpighian tubules are degraded, spiroplasma utilizes nutrients derived from the sarcolemma of the muscle cells. Consequently, the effects of the insecticide are likely enhanced, accelerating the death of the insects, which are physiologically weakened by the pathogen.

On each evaluation date, it was observed that the cumulative mortality of the leafhoppers increased over the days following the infestation of the plants. However, when comparing the evaluation dates (0, 5, 10, 15, and 20 days), a reduction in mortality was noted as the leafhoppers were exposed to older plants, indicating a decline over time after the seed treatment (Figure 3). This may be related to the high solubility of neonicotinoids, which leads to the rapid absorption and metabolism of these compounds by the seed after planting. In other words, lower concentrations in the new tissues of the plant reduce the product’s efficacy [34,35,36].

The infective *D. maidis* adults requires a feeding period of around 1 h to transmit the spiroplasma [12]. Therefore, understanding the physicochemical properties of insecticides is crucial, as active ingredients with rapid action that disrupt or paralyze insect feeding can help reduce the risk of pathogen transmission to the plant. The insecticides evaluated in this study were selected due to their systemic action and for targeting the insect’s nervous system during feeding. Those in the neonicotinoid and pyrethroid chemical groups induce paralysis and death through a knockdown effect [37]. Pyrethroids act on sodium channels, causing them to remain open for a longer duration, resulting in insect death due to hyperexcitability. Also, as neurotoxic compounds, neonicotinoids are acetylcholine agonists that bind to nicotinic receptors, opening sodium channels. Nevertheless, the molecules are not immediately degraded, leading to hyperexcitation of the nervous system and subsequent insect death [37]. In contrast, carbamates paralyze nerves and muscles by inhibiting the acetylcholinesterase enzyme [38,39]. The molecular analyses of the plants at time 0 allowed us to verify whether pathogen transmission occurred, relate the mortality rate to disease transmission, and validate the assessment scales used represented in Figure 5. Imidacloprid/thiodicarb was the insecticide that provided 100% protection by preventing spiroplasma transmission. The insecticides thiamethoxam and thiophanate-methyl + fipronil/thiamethoxam reduced the transmission rate of spiroplasma to 20% and 10% of the plants, respectively (Figure 6).

We hypothesize that these products, in addition to showing high initial mortality in leafhoppers, reduce the feeding time of the insects, contributing to the decrease in spiroplasma transmission. However, the products were only effective in preventing symptoms up to the V2 stage. This result is consistent with expectations for neonicotinoids, carbamate, and pyrethroid, which typically have a residual efficacy period from 15 to 21 days after plant emergence, depending on the soil, climatic conditions, and type of application [13,40,41]. After V2, however, with the gradual reduction of the residual effect of the products, seed insecticide treatments were not effective, and symptoms of CSS were observed in the plants, leading to yield losses.

CDSC have a lengthy developmental cycle in plants. The pathogens involved (phytoplasma and spiroplasma) require several months to multiply and reach a titer within the plant that can lead to significant economic losses. Consequently, the earlier the plants become infected, the greater the potential losses are in production [42]. In this study, we observed that the earlier the exposure of plants was to infective leafhoppers, the greater the reduction was in productivity-related parameters (Figure 6A,B), due to the longer period the pathogen had to multiply within the plant, allowing for phloem colonization and interrupting nutrient translocation [40,41]. Thus, the more effective the seed treatment was, the lower the impact was on productivity parameters [43].

The efficiency of pathogen transmission by the vector insect is not necessarily the most relevant factor from an epidemiological perspective. Pathogen transmission and disease spread depend on various factors, such as the infectivity rate of the insect population and its abundance, as well as other characteristics like biotic potential, geographic distribution, dispersal capability, and the period during which host plants are most susceptible [44]. This suggests that it is important to avoid high densities of leafhoppers in corn fields, especially during the early growth stages (VE-V4); however, this protection is discontinued at later growth phases. Delayed control of the insect vector, leading to population growth, can result in significant yield losses due to CSDC. Seed treatment, combined with good agricultural practices, is necessary to reduce the population density of leafhoppers in the field, ensuring a lower risk of the occurrence of CSDC. Finally, the higher mortality of infective leafhoppers carrying spiroplasma, compared to healthy leafhoppers, when exposed to chemical products is a novel finding that may open new research directions for future studies.

## 5. Conclusions

The seed treatments were effective in the initial evaluation, indicating that their protective effect persisted for approximately 5 days following the emergence of the second fully expanded leaf, or roughly 11 days after sowing. Among the evaluated products, the combination of imidacloprid and thiodicarb proved to be the most effective, providing complete (100%) protection against *Spiroplasma* infection. In contrast, thiamethoxam and the mixture of thiophanate-methyl + fipronil/thiamethoxam conferred partial protection, preventing the expression of *Spiroplasma*-associated symptoms in 80% and 90% of the plants, respectively. This has important implications for field experiments, especially for managing the insect stunting complex (CSDC). It is important to consider the possibility of reinfection and migration of healthy leafhoppers that survived seed treatment, which may require additional control measures to ensure continued protection of corn plants. Insect survival in response to control tactics may be altered, and therefore the recommendations need to be revisited.

## Figures and Tables

**Figure 1 insects-16-00713-f001:**
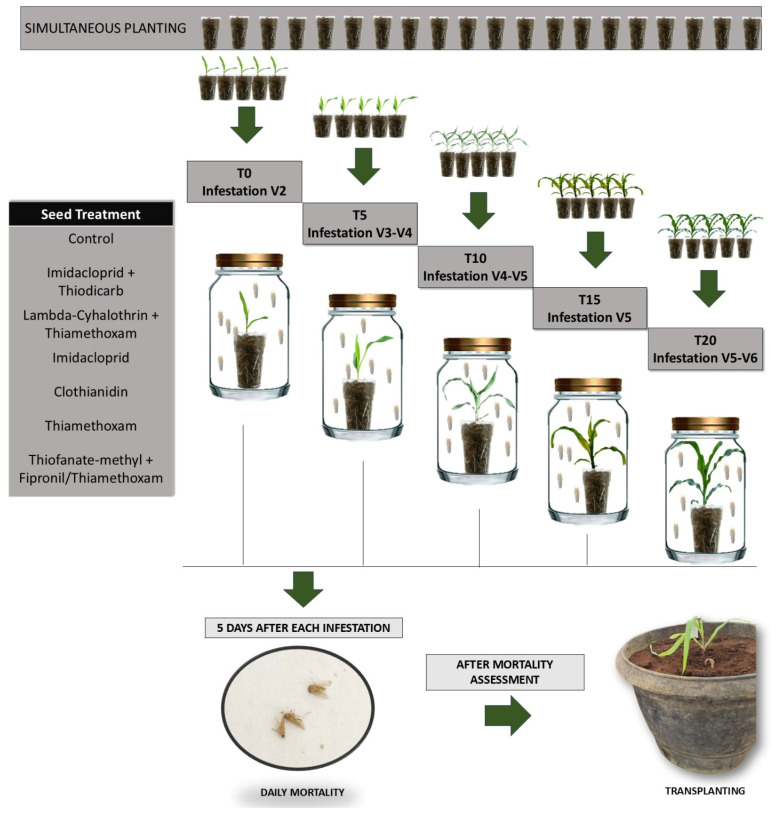
Simplified schematic view of the methodology: sow of treated seeds, leafhopper infestation, mortality assessment, and maize plants exposed to leafhoppers transplanted in the greenhouse.

**Figure 2 insects-16-00713-f002:**
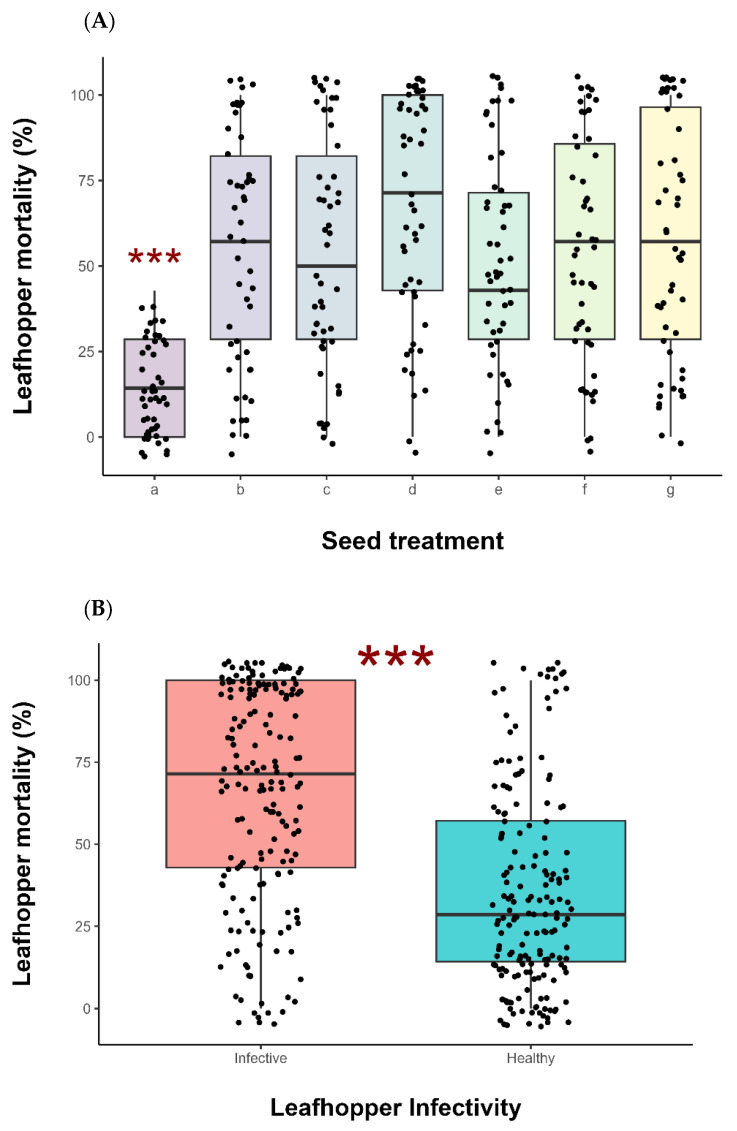
(**A**) Median mortality of all *D. maidis* exposed to insecticides seed treatment. Insecticide seed treatment *x*-axis legend: a = water (=control), b = imidacloprid/thiodicarb, c = Lambda-Cyhalothrin/thiamethoxam, d = imidacloprid, e = Clothianidin, f = Thiamethoxan, and g = thiophanate-methyl + fipronil/thiamethoxam. *** indicates the bar that differs from the others in terms of *p* ≤ 0.001. (**B**) Median mortality of infective and healthy *D. maidis* individuals exposed to seed treatment. *** indicates the bar that differs from the others in terms of *p* ≤ 0.001.

**Figure 3 insects-16-00713-f003:**
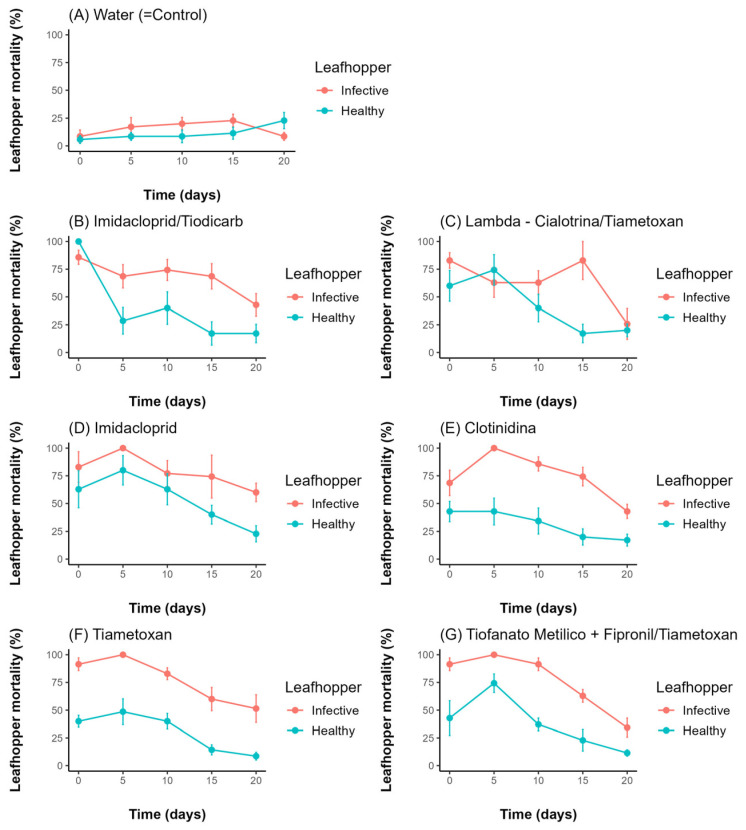
Total mortality (mean ± SEM) of infective and healthy *D. maidis* individuals over time (days) following exposure to control (water) and six insecticide seed treatments. After seed treatment, there were five infestations (one per time), corresponding to 0, 5, 10, 15, and 20 days after the appearance of the second expanded leaf.

**Figure 4 insects-16-00713-f004:**
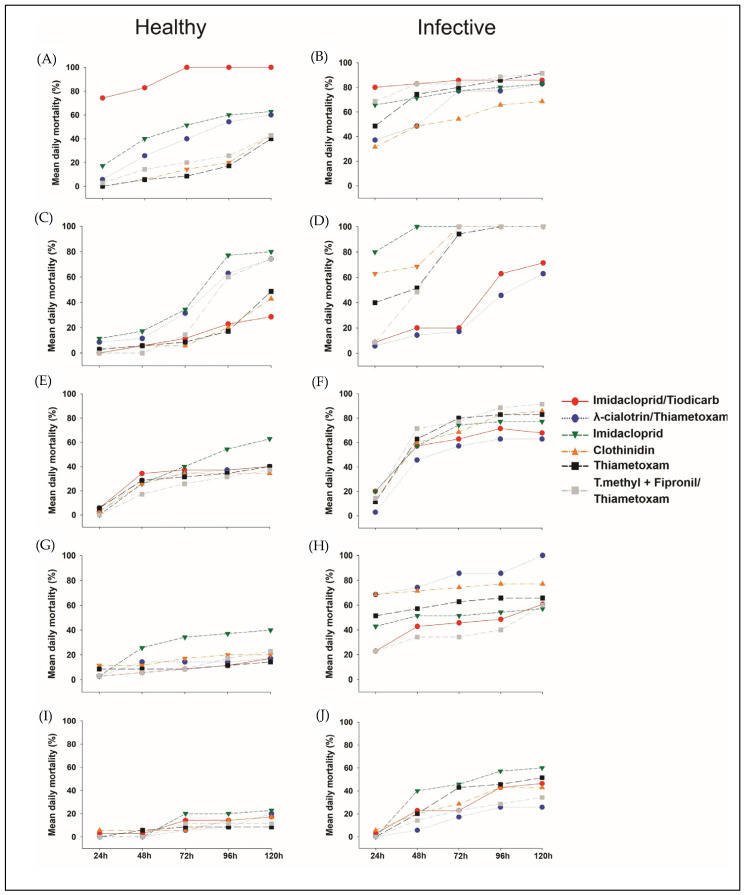
Leafhopper mean daily mortality (%) after different insecticide treatment during five consecutive days (24, 48, 72, 96, and 120 h). Legend: (**A**) = healthy leafhoppers at time 0, (**C**) = healthy leafhoppers at time 5, (**E**) = healthy leafhoppers at time 19, (**G**) = healthy leafhoppers at time 15, and (**I**) = healthy leafhoppers at 20 days after infestation. (**B**) = infective leafhoppers at time 0, (**D**) = infective leafhoppers at time 5, (**F**) = infective leafhoppers at time 10, (**H**) = infective leafhoppers at time 15, and (**J**) infective leafhoppers at 20 days after infestation.

**Figure 5 insects-16-00713-f005:**
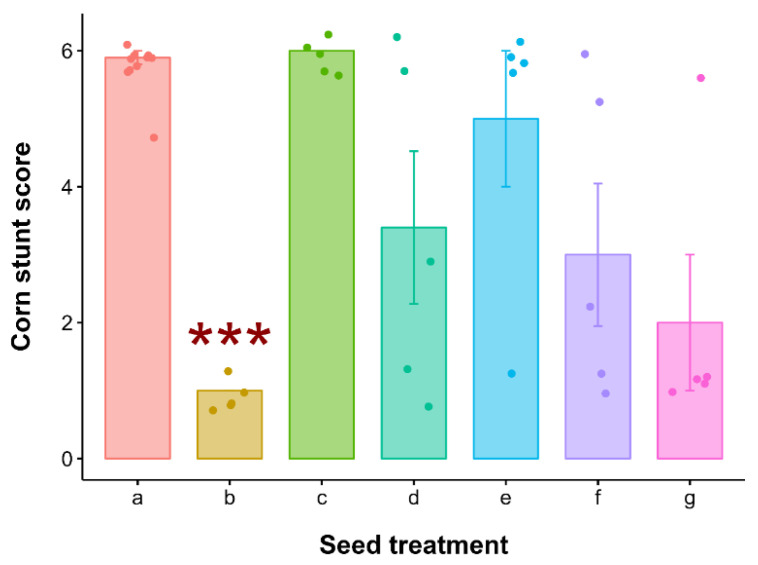
Corn stunt score (mean ± SEM) by seed treatments at time 0 (after 5 days of seed insecticide application). Corn stunt was evaluated at the R4 growth stage using the visual symptom rating scale from 1 to 6 (Silva et al., 2003 [18]). Seed treatment *x*-axis legend: a = water (=control), b = imidacloprid/thiodicarb, c = Lambda-Cyhalothrin/thiamethoxam, d = imidacloprid, e = Clothianidin, f = Thiamethoxan, and g = thiophanate-methyl + fipronil/thiamethoxam. Treatments are represented by different colors. *** indicates the bar that differs from the others in terms of *p* ≤ 0.001 by the Dunn a posteriori non-parametric test.

**Figure 6 insects-16-00713-f006:**
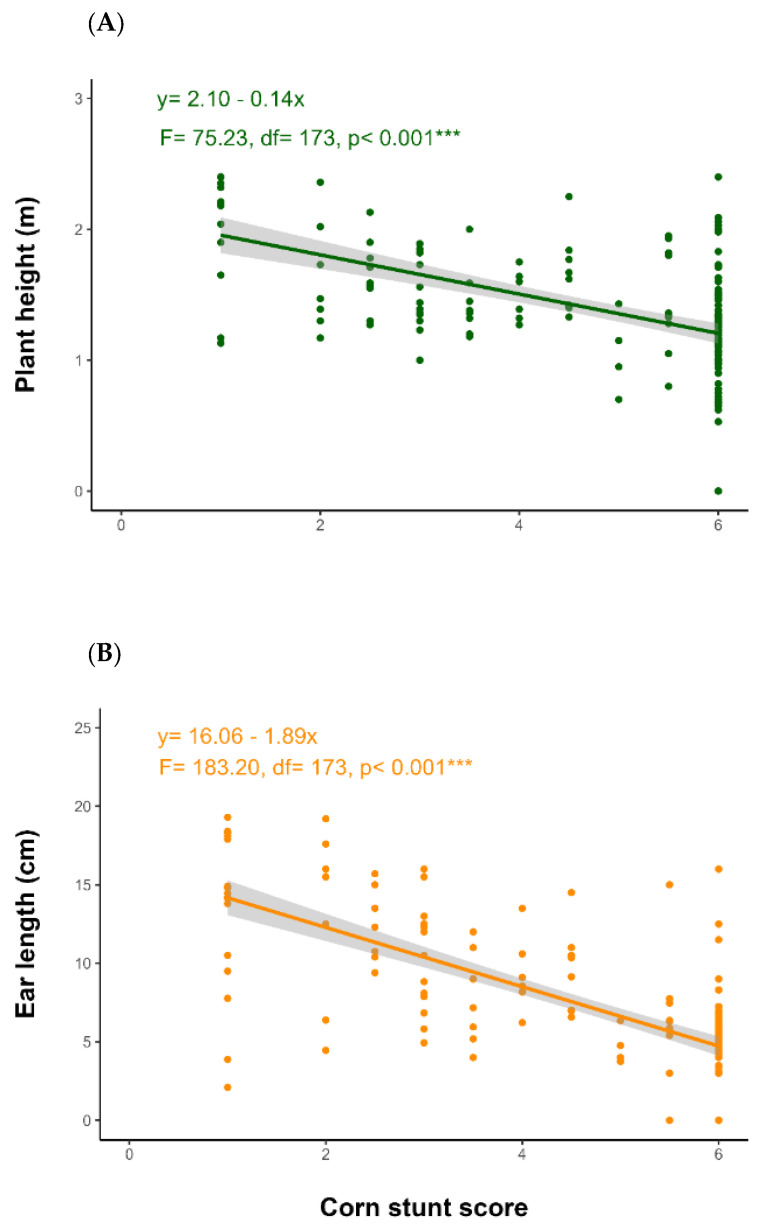
Effect of corn stunt severity on maize height and ear lengths in the greenhouse evaluations. (**A**) Plant height (m) and corn stunt score, (**B**) ear length (m) and corn stunt score. The corn stunt score evaluated using the visual symptom rating scale from 1 to 6 (Silva et al., 2003) [18]. *** indicates the bar that differs from the others in terms of *p* ≤ 0.001 by the Dunn a posteriori non-parametric test.

**Table 1 insects-16-00713-t001:** Trade name, active ingredient, chemical group, concentration, and dose of the insecticides used in the seed treatment of maize seeds for the control of *Dalbulus maidis* adults in Sete Lagoas/MG, Brazil, between September and December 2023.

Trade Name ^(1)^	Active Ingredient	Chemical Group	Concentration ^(2)^	Dose ^(3)^
Control	Water	----	-----	-----
Nuprid Star FS	Imidacloprid + tiodicarb	Neonicotinoid + carbamate	150 + 450	1.75
Cruiser Opti	Lambda Cyhalothrin + thiamethoxam	Pyrethroid + neonicotinoid	37.5 + 210	1
Ouro fino	Imidacloprid	Neonicotinoid	350	0.08
Poncho	Clothianidin	Neonicotinoid	660	0.1
Cruiser 600 FS	Tiametoxam	Neonicotinoid	600	0.5
Standak Top*+ Impar BR	Thiophanate methyl + fipronil/thiamethoxam	Pyrethroid + neonicotinoid	225 + 250/350	1/0.08

^(1)^ Registered in the Ministry of Agriculture, Livestock and Supply (Mapa) for *Dalbulus maidis*, with the exception of Standak top*. ^(2)^ Concentration (g a.i. L^−1^ trade inseticide), ^(3)^ dose trade insecticide L 100 Kg^−1^ of seed.

## Data Availability

The original contributions presented in the study are included in the article/Appendix A. Further inquiries can be directed to the corresponding authors.

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
