# Peer review of "What Is the Relationship Between Efficacy of Seed Treatment with Insecticides Against Dalbulus maidis (Delong and Wolcott) (Hemiptera: Cicadellidae) Healthy and Infected with Spiroplasm in the Corn Stunt Control?"

_insects, 2025, doi:10.3390/insects16070713_

Round 1

Reviewer 1 Report

Comments and Suggestions for Authors

Overall, the merit is good; the manuscript is descriptive and well-directed, with an appropriate style and information that is truly valuable. However few points below could enrich substantially this submission:
1. Scientific name must be italicized throughout the text
2. Last paragraph of the introduction could be improved by adding data on the Americas regarding diseases transmitted by this leafhopper
3. Select other color for Fig 1 which is too bright 
4. Fig 2 is blurry, update in better quality
5. Fig. 3 is too blurry and hard to read due to font size. Update it!
6. Conclusion could get a substantial improvement by adding data obtained in experiments, extend it

Reviewer 2 Report

Comments and Suggestions for Authors

Review

This is a significant paper in the field of leafhoppers control, and through this insects, in Spiroplasma prevention.

The manuscript is well designed, I especially enjoyed the methods part, the authors made significant effort to test insecticide mortality on infected and healthy leafhoppers. Figures are well designed, however for several one (i.e. Figure 3 and 4) the letter font inside the figures must be increased.

I consider that PCR figures are no more important for main text, these can be placed in supplementary materials.

I have no negative comments, again, I do appreciated the effort made in this research.

I suggest minor revision, and acceptance of this paper.

Reviewer 3 Report

Comments and Suggestions for Authors

Article: What is the relationship between efficacy of insecticides seed treatment against Dalbulus maidis (Delong & Wolcott) (Hemiptera: Cicadellidae) healthy and infected with spiroplam in the corn stunt control?

General comments:

  1. The scientific names of insects such as D. maidis, plants such as Zea mays, and pathogens such as S. kunkelii should always be italicized. Please check the formatting across all pages of the manuscript.
  2. The differences between each plant growth stage from V2 to V6 should be clearly explained.
  3. The manuscript refers to the same insect species using various terms, including D. maidis, "leafhopper", and "corn leafhopper". For consistency and to avoid reader confusion, choose only one term and use it consistently throughout the text.

Specific comments:

  1. Article
    • Line 2: Use the term “seed insecticides” instead of “insecticides seed” to conform to correct grammatical structure.
  2. Keywords
    • Line 40:
      • Remove “chemical control” as it is redundant with “insecticides”.
      • Correct the scientific name formatting: capitalize the first letter of the genus and italicize the whole name — Zea mays.
      • Consider adding important keywords such as “corn leafhopper” or Dalbulus maidis.
  3. Materials and Methods
    • Line 123: Check for typographical errors.
    • Line 128: Ensure accuracy in stating the number: 2,450.
    • Line 138: Provide additional explanation for the term “(R4) maize growth stage”.
  4. Results
    • Include a comparative analysis of the efficacy of all insecticides used, ranking them from most to least effective.
    • Line 251: Check the placement of the full stop.
  5. Discussion
    • Line 273: Verify the correctness of the abbreviation “CDC”.
    • Add references comparing findings with other studies, particularly regarding the relationship between the infective group and the effect of insecticides.
  6. References
  • Please note that Reference 45 is missing from the reference list. Kindly check the in-text citation and ensure the reference is included and formatted correctly.

Author Response

Resposta em anexo
